# Combined Solid-State LiDAR and Fluorescence Photogrammetry Imaging to Determine Uranyl Mineral Distribution in a Legacy Uranium Mine

**DOI:** 10.3390/s25072094

**Published:** 2025-03-27

**Authors:** Thomas B. Scott, Ewan Woodbridge, Yannick Verbelen, Matthew Ryan Tucker, Lingteng Kong, Adel El-Turke, David Megson-Smith, Russell Malchow, Pamela C. Burnley

**Affiliations:** 1Interface Analysis Centre, H.H. Wills Physics Laboratory, University of Bristol, Bristol BS8 1TL, UK; ewan.woodbridge.2018@bristol.ac.uk (E.W.); lingteng.kong@bristol.ac.uk (L.K.); a.el-turki@bristol.ac.uk (A.E.-T.);; 2Hot Robotics (NNUF-HR), University of Bristol, Bristol BS8 1TL, UK; 3Department of Geoscience, University of Nevada, Las Vegas, NV 89154-4010, USA; pamela.burnley@unlv.edu

**Keywords:** uranyl, minerals, fluorescence, surveying, mapping, LiDAR, legacy mine, uranium

## Abstract

Determining the presence and abundance of uranium mineralization at legacy mine sites is important both for responsible environmental management and potential resource recovery. Technologies that can make such determinations quickly and at low costs are highly desirable. The current work focuses on demonstrating the use of simple handheld commercial-off-the-shelf (COTS) devices for rapidly determining the presence and distribution of uranyl minerals within an abandoned copper–uranium mine. Specifically, this work demonstrates the use of a COTS iPhone 13 Pro smartphone with an inbuilt solid-state LiDAR (laser) scanner in combination with a handheld LED-based UV torch to conduct a rapid fluorescence imaging photogrammetry survey aimed at rapidly determining the distribution of uranyl minerals within an abandoned copper–uranium mine in the Sierra Ancha Wilderness Area, Gila County, Arizona, USA. Such a simple methodology, presented herein, can be used to quickly determine the distribution of uranyl minerals on exposed surfaces within the underground workings and provide an indication of the presence of primary uranium ore minerals buried within the surrounding rock.

## 1. Introduction

In many countries, active mining operations are required to conform to environmental regulations, which often entails actively accounting for and managing arising tailings, other wastes, and effluents to ensure that surface and groundwater sources are not contaminated with potentially harmful heavy metals and other toxins. At the end of the operational lifetime of such a mining facility, it is the duty of the site owner to ensure that the workings are made safe and that any environmental restoration has been adequately implemented. However, legacy mine sites, particularly those abandoned before modern environmental regulations were established, often lack adequate containment measures, making them persistent sources of environmental pollution. Such sites should not pose an ongoing source of risk to the surrounding environment, wildlife, and local population [1]. Historically, such responsible custodianship has not always been followed, partly due to a lack of regulatory oversight or enforcement. This means that, across the world, there are many mining sites that have not been suitably managed beyond the end of their working lifetimes. Possibly the most infamous example is the Rio Tinto mining area in the Iberian Pyrite Belt, which constitutes an extreme case of pollution by acid mine drainage [2] arising from large-scale mining activities in the 18th century. The ongoing contamination has led to severe ecological degradation, with the Rio Tinto river exhibiting extreme acidity and elevated concentrations of heavy metals. The poor management of the mine workings has rendered large sections of the river largely uninhabitable for most aquatic life. Despite these challenges, remediation efforts have been undertaken, including the implementation of water treatment systems and controlled flooding of abandoned workings to reduce acid generation. Whether conducted above ground or underground, mining can release toxic metals into the air and water. Meteoric water or groundwater contacting the mine workings or surface tailings can accumulate harmful concentrations of heavy metals through dissolution reactions. This contaminated water, if left untreated, can potentially pollute the area directly surrounding the mine and beyond, including sources of drinking water for local human populations [3].

For many mining operations where metals such as iron, copper, or gold are being extracted, there are accessory minerals that contain low levels of radioactive isotopes that can become concentrated in mine tailings and subsequently dissolved into waters moving through the site along with other toxic heavy metals. Radionuclides can also be released in dust that is generated during physical–mechanical mining processes or from wind-blown tailings piles and dried-out ponds. The likelihood of environmental pollution is substantially increased if the radioactive minerals have been separated into tailings that have not subsequently been well protected from leaching by meteoric waters. Once radionuclides are released into an ecosystem, they can accumulate in plants and subsequently ascend the levels of the food chain [4]. Radioactive contamination is therefore a major safety issue that needs to be actively managed both during the operation of a mine and following its closure.

Uranium is the most notorious radioelement associated with mine wastes. Uranium is actively mined in its own right for nuclear fuel production, with ores containing high concentrations of minerals such as uraninite (pitchblende) (U_3_O_8_). With a global nuclear renaissance occurring in response to divestment from fossil fuels, there is current active interest in mining uranium, including the revisiting of old mines and legacy prospects to evaluate them for residual uranium resources. Equally, uranium can occur as an accessory element in other precious metal ore bodies such as copper, tin, and gold, appearing in minerals such as (meta)torbernite (Cu(UO_2_)_2_(PO_4_)_2_ · 8 H_2_O). The most notable co-occurrence of uranium and thorium for mining is associated with the mining of Rare Earth Elements (REEs), where they are notable elemental components of the monazite (phosphate) mineral ores and therefore provide an unavoidable hazard. For example, monazite mining has been strictly regulated by the US and China, who are the major global REE producers [5,6,7].

Today, there are many thousands of legacy mine sites across the world that have been inadequately managed or monitored beyond their period of operation. The US Department of Agriculture’s Forest Service and the Department of the Interior’s Bureau of Land Management (BLM) have identified at least 140,000 abandoned hardrock mine features, such as tunnels, on their controlled land. Of these, it is estimated that about 50% may pose physical safety hazards and about 15% may pose environmental hazards to human health or wildlife [8]. Additionally, the agency officials have estimated that there could be an additional 390,000 abandoned mine features on federal land that they have not captured in their databases [8]. For example, in the Navajo Nation, Arizona, some 523 sites were explored and exploited for uranium between 1944 and 1984, spanning the period of the Cold War (1947–1986), with some 30 million tons of uranium ore extracted over this period [9]. Today, there is a substantial Environmental Protection Agency (EPA)-funded program, valued at USD 1.7 billion, focused on surveying and then remediating 230 abandoned uranium mines on Navajo lands, addressing the sites that pose the highest risk of radiation exposure to the Navajo people [10]. The challenge that is present in the US is represented in numerous countries around the world, including Canada, Australia, France, Czech Republic, Kazakhstan, Romania, South Africa, and Russia. This challenge and opportunity is therefore a global concern.

Given the substantial task of reclaiming abandoned mine lands, suitable technologies must be developed and applied to increase the speed and efficiency of environmental surveying while ideally reducing costs. The use of remotely operated robotics such as unmanned aerial vehicles (UAVs) is becoming increasingly popular for environmental surveys of legacy mine sites [11,12]. These technologies continue to advance, and the costs associated with their deployment are expected to decrease, making them an even more viable option for large-scale environmental monitoring and reclamation projects.

Whilst UAVs are ideally suited to open-air surveys, they face significant limitations when it comes to operating in confined indoor spaces, such as underground mines. These environments present unique challenges, including limited GPS access, complex layouts, water, and obstacles that hinder the safe operation of UAVs. Additionally, the current indoor drone technology is constrained by short flight times, often less than 10 min [13]. As a result, alternative methods, such as ground-based robotic systems or traditional manual surveys, are still required to effectively gather data in these challenging environments.

The distribution of residual radioactive minerals within a mining site is typically determined using radiometric measurement instruments. These devices are used to measure the gamma photon flux being emitted from exposed surfaces around the mine. These relatively expensive instruments are sometimes paired with devices that determine the physical location within a space, such as the GPS and/or by the use of simultaneous localization and mapping (SLAM) [14] using cameras or laser scanning. While radiometric techniques can effectively identify radioactivity and the specific radioactive isotopes that are present, they are relatively slow and cumbersome for the mapping of naturally occurring radioactive materials (NORMs) due to the need for large volumes of detector materials and/or extended counting times to obtain statistically reliable data. Additionally, gamma spectrometry techniques are unable to ascertain the chemical state of the observed materials. This limitation is particularly pronounced in mining environments, where the radioactivity levels are very low compared to civil nuclear sites and facilities. Furthermore, the presence of radon gas, a common byproduct in underground and some open-pit mines, can cause detector contamination from gaseous ingress, rendering the device temporarily unusable.

For uranium minerals, determining the chemical state is particularly valuable because uranyl minerals, which are those containing uranium in its 6+ valence state, are considered as being highly susceptible to dissolution relative to uranium 4+ minerals, which are considered to be relatively insoluble [15]. Uranyl (U6+) minerals are more susceptible to dissolution in oxidizing fluids such as water versus primary U4+ minerals because they have high solubility due to uranyl ions readily forming stable aqua-complexes with groundwater anions. Uranyl minerals frequently originate as secondary minerals formed from tetravalent ‘primary’ precursors such as uraninite (UO_2_) and coffinite (USiO_4_) but can also mineralize from the processes of evaporation and precipitation. Uranyl minerals are considered to pose a greater environmental pollution risk; hence, understanding whether an abandoned mine contains uranyl minerals is extremely important when assessing the potential pollution risk.

Ultraviolet (UV) fluorescence imaging can assist with the rapid visualization of the distribution of minerals exhibiting fluorescence, including uranyl-bearing minerals. UV light at 365 nm stimulates fluorescence in a variety of different materials due to the absorption of UV radiation, which excites electrons to higher energy states. When these electrons return to their ground state, they emit visible light as fluorescence [16]. The specific wavelength and intensity of the emitted fluorescence can be used to identify the mineral species, detect variations in crystal chemistry, and assess the quality or purity of the material. Additionally, fluorescence can reveal insights into the environmental conditions under which the mineral formed, such as temperature and pressure, as well as any post-formation alterations [16,17,18]. In a mine site setting, the occurrence of fluorescing minerals is sparse because most rock-forming minerals do not exhibit this behavior. This makes rapid identification and localization of minerals or ores relatively simple.

Table 1 shows the majority of the common uranyl minerals that are observed to exhibit characteristic yellow–green fluorescence when illuminated by a UV source. Figure 1, adapted from [19], shows that this property can potentially be exploited, especially in dark underground workings, to quickly show if such minerals are present on exposed rock surfaces and, if so, map their exact location and distribution.

The current work focuses on demonstrating the use of simple handheld COTS devices for rapidly determining the presence and distribution of uranyl minerals within an abandoned copper–uranium mine. Specifically, this work demonstrates the use of a COTS iPhone 13 Pro smartphone with an inbuilt solid-state LiDAR (laser) scanner in combination with a handheld LED-based UV torch to conduct a rapid fluorescence imaging photogrammetry survey aimed at rapidly determining the distribution of uranyl minerals within an abandoned copper–uranium mine in the Sierra Ancha Wilderness Area, Gila County, Arizona. Such a simple methodology, presented herein, can be used to quickly determine the distribution and abundance of uranyl minerals within underground workings and provide an indication of the presence of primary uranium ore minerals buried within the surrounding rock.

## 2. Materials and Methods

### 2.1. Lithological Location

The measurements presented herein were captured inside the abandoned Donna Lee copper–uranium mine at the southernmost extent of the Sierra Ancha Wilderness Area, Gila County, Arizona (33°46′17″ N, 110°51′43″ W). Based on historic records, mines in the area were known to contain accessory uranium mineralization. Given this, and the understanding of the regional geology, it was inferred that secondary uranium-bearing minerals, likely in the form of uranyl, would also be present.

Located at an altitude of 1395 m and accessed via a narrow miner track that wraps around the mountainside, as presented in Figure 2, the mine adit was a simple horizontal drive cutting approximately 24 m into a siltstone horizon of the Dripping Springs Quartzite (DSQ). The DSQ, which regionally hosts numerous uranium-bearing vein deposits and associated mines, is part of the Mesoproterozoic Apache group (1328 Ma to 1256 Ma), a sequence of clastic sedimentary rocks, dolomitic limestone, and basalt flows that rests unconformably on early Proterozoic crystalline basement (Vishnu Ruin Granite). The uranium mineralization is thought to be associated with incompatible fluids developing from the intrusion of diabase dikes and subsequently from deposited Precambrian sediments around 1050 Ma [23]. The fluids are thought to have migrated away from the diabase dikes via high-angle, north-northeast, and west-northwest trending joints and deposited uranium minerals when they reached the siltstone member of the DSQ, which presented appropriately low oxidation–reduction potential (Eh) conditions for precipitation [24]. Primary mineralization identified at the Donna Lee mine includes pyrite (FeS_2_), chalcopyrite (CuFeS_2_), and calcite (CaCO_3_). Poorly crystallized uraninite, although not identified, is likely present as well as it is the primary form of uranium mineralization in the DSQ [24]. Secondary minerals, which occur as efflorescent coatings on the surfaces of fractures and bedding planes at the Donna Lee mine, reportedly include metatorbernite (Cu(UO_2_)_2_(PO_4_)_2_ · 8 H_2_O), limonite (FeO(OH) · n H_2_O), malachite (Cu_2_CO_3_(OH)_2_), chalcanthite (CuSO_4_ · 5 H_2_O), and gypsum (CaSO_4_ · 2 H_2_O), with the constituent sulfur and phosphorus likely derived from oxidative dissolution of sulfide minerals [24].

### 2.2. Data Collection

#### 2.2.1. Initial Scout Survey

The mine adit was initially explored using an expert 2-man survey team, suitably equipped with caving equipment (helmets, head torches, and oxygen monitors) as well as Hamamatsu Photonics gamma spectrometers, Iwata City, Japan, Radhound Geiger–Müller counters, and Tracerco PED+ dosimeters, Sheffield, UK. The scout survey gauged the ambient radioactivity within the mine as well as its internal layout and structural stability.

#### 2.2.2. Fluorescence Survey

The subsequent fluorescence survey was carried out by using a single person carrying a Klear Concepts UV light source and coincident Apple iPhone 13 Pro smartphone, CA, USA, as demonstrated by Figure 3. The iPhone 13 Pro, shown in Figure 4, has an incorporated solid-state LiDAR Scanner, 3 RGB cameras, is IP68 rated, operational up to 35 °C, and well suited to low-light acquisitions [25]. Data from the LiDAR scanner and the cameras are combined using commercial software to create a 3-dimensional model of the space.

The emission wavelength of the LiDAR is approximately 940 nm, but the precise specifications of the LiDAR have not been released by Apple. However, the performance of the sensor has been extensively evaluated. Other studies [26,27,28] have confirmed that the LiDAR system in the iPad Pro uses direct Time of Flight technology by shining a Vertical Cavity Surface Emitting Laser (VSCEL), and measuring the reflected beams using an array of Single-Photon Avalanche Detectors (SPADs) [28]. Due to its low cost and weight relative to commercial LiDAR systems, the iPhone is increasingly being applied to novel mapping tasks. Its performance has been evaluated for geoscience applications, mapping large cliff faces (with dimensions of 130 × 15 × 10 m) to an accuracy of ±10 cm [28]. Evaluations for indoor mapping have found an improved spatial accuracy to 3 cm when modeling a room 22 × 6 m in size [29].

A KLEAR Concepts UV LED blacklight, shown in Figure 4, was used for illumination of the mine adit. Emitting at a peak wavelength of 395 nm with 2 W power and emitting from across a 78 mm diameter area (100 LEDs), the waterproof unit weighed only 399 g, making it ideal for handheld underground deployment. The UV blacklight was found to have a wavelength of emission that extended slightly into the visible, and this weak purple/blue illumination was found to be sufficient for the operative to safely conduct the survey.

**Figure 4 sensors-25-02094-f004:**
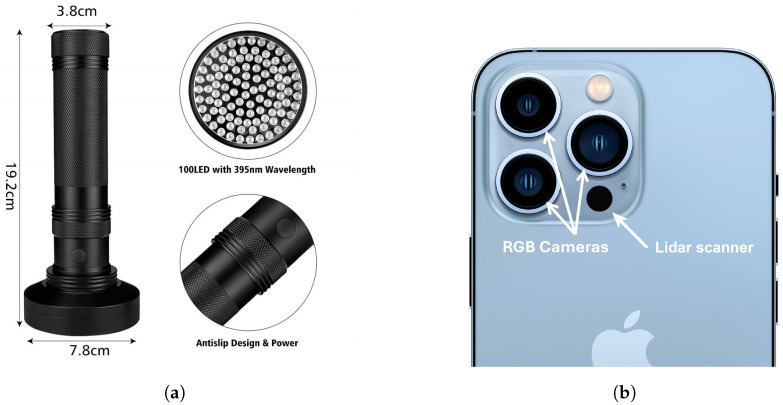
(**a**) KLEAR UV LED Torch. (**b**) iPhone 13 Pro frontal camera and LiDAR array, adapted from [30].

The UV blacklight source was held immediately above the smartphone, pointing directly forward such that the area of illumination was aligned with the center of the field of view of the phone’s camera and LiDAR.

The survey was collected by gradually progressing into the adit, sweeping the camera phone and light source in a circular motion to enable progressive 3D capture of the physical structure, as well as concurrently capturing an RGB orthophoto overlay via the 3 RGB cameras. When the end of the adit was reached, the operative turned back towards the mine entrance and the survey was slowly continued until the adit entrance was exited. At the adit entrance, sunlight provided good illumination, but, within ≈4 to 5 m, natural light was sufficiently dimmed that the data capture was carried out in darkness, thereby improving the efficacy of the setup for fluorescence imaging. No other light sources were used in the survey. The operative captured the data without the use of a head torch, and UV-protective glasses were worn to protect from strong UV reflection. The survey itself took approximately 20 min, and better than 95% capture of the adit was achieved in this period.

#### 2.2.3. 3D Rendering Software Canvas

Canvas is an Apple IOS mobile application that utilizes the iPhone’s LiDAR sensor to create detailed 3D scans of physical spaces. It first prompts the user to move their device around the space they wish to capture, assisting in comprehensive coverage of all surfaces. During this data capture phase, Canvas continuously collects depth data from the LiDAR sensor, which are combined with visual data from the RGB cameras to build a detailed point cloud representation of the environment with photographic overlay that can be processed on the device to generate a preliminary 3D model. Data can also be uploaded to Canvas’s cloud-processing servers to generate a higher-resolution model.

Once the raw scan data are uploaded to Canvas’s cloud-based processing service, the platform employs advanced computational algorithms to enhance and refine the initial point cloud captured by the iPhone’s LiDAR sensor. The cloud-based processing first performs noise reduction and error correction to ensure that the point cloud accurately represents the scanned environment. Next, the data undergo a meshing process, where the points are connected to form a continuous 3D surface or mesh. The processing pipeline also applies texture mapping, using the RGB data captured during the scan to overlay realistic textures onto the 3D model, creating a highly detailed and visually accurate representation of the space. Occipital Ltd., the developer of the Canvas software, states that the models used are spatially accurate to within roughly 2% [31] For the mine survey herein, this equates to approximately 3 to 4 cm.

#### 2.2.4. Collection of Ground Truthing Samples

The selection of uranyl-bearing mineral samples was guided by a combination of radiometric and visual indicators. Samples that exhibited a strong yellow–green fluorescence under UV illumination were assessed for strength of emitted radioactivity using a Geiger–Müller (GM) counter. Priority was given to samples displaying both elevated radiation counts and strong fluorescence as these features are indicative of uranyl mineralization. An example photograph is shown in Figure 5.

#### 2.2.5. Laboratory Analysis of Mine Samples

Field Emission Gun Scanning Electron Microscopy (FEG-SEM) combined with energy-dispersive X-ray spectroscopy (EDX) and Raman spectroscopy were employed to achieve a comprehensive analysis of the collected mine samples (Figure 6). FEG-SEM provides high-resolution imaging, enabling precise observation of a sample’s surface morphology down to the nanoscale. EDX facilitates the identification and quantification of the elemental composition by detecting characteristic X-rays emitted during electron bombardment.

Raman spectroscopy complements these techniques by providing molecular-level information through the analysis of bond vibrations (vibrational modes) accessed via Raman scattered light arising from monochromatic laser illumination of a sample. Measurements utilized a red laser (633 nm), and the resulting spectrum was analyzed to identify molecular structures and phases. Together, these analytical techniques enabled a detailed determination of the mineralogical composition.

The Raman spectral library RRUFF Project (http://rruff.info/index.php (accessed on 17 January 2025)) [32] was used to to compare the recorded Raman spectra with well-documented reference spectra of known minerals to provide a mineral identification.

### 2.3. In-Mine Determination of Uranyl Mineral Distribution

To analyze the in-mine distribution of uranyl minerals, from the captured 3D data, the color of a representative fluorescing pixel in the image mesh can be chosen and the RGB value extracted. However, differences in lighting and the amount of fluorescence mean that not all fluorescing pixels will be exactly the same color. To ensure that a wider range of pixels will be selected, the distance within the RGB color space can be calculated. The RGB value of each pixel is treated as a 3-dimensional vector and the Euclidean distance between the pixels and the representative RGB calculated.

## 3. Results

### 3.1. Canvas Renders

The results of the solid-state LiDAR scanning and combined fluorescence imaging are shown in Figure 7, Figure 8, Figure 9, Figure 10 and Figure 11. The laser scan data show almost complete coverage of the inside of the mine adit. There were some small areas that were not covered, accounting for less than 2%. These blindspot areas, where there is a gap in the recorded 3D mesh, were attributed to occlusions caused by overhang of the rock and or incomplete survey sweeping of the smartphone in both the inward and outward directions.

The laser scan was able to capture all the major features within the mine and was able to verify that the mine tunnel ran 28 m into the cliff face, with a ceiling height of approximately 2 m and a tunnel width of approximately 1.4 m. The physical structures in the rock at the entrance to the main tunnel were physically measured at the time of the survey and subsequently compared with the same structures as they were captured within the 3D model. This comparison confirmed that the spatial accuracy of the solid-state line was on the order of 3 to 4 cm, which correlates with the manufacturer specifications for the device. The mine adit was even evidenced to be sloping gently upwards from the entrance, which is a standard feature that is common to most mines, which promotes drainage of seeping waters out of the mine (Figure 7).

The inspection of the orthophoto overlay on the 3D mesh showed extremely good correlation with the photographs taken by the entrance to the mine. This is shown in Figure 9 via a photograph of the main entrance compared against a 3D orthophoto reconstruction based on RGB photogrammetry information. The primary difference between the two images relates to the shading of the corresponding scene. The photograph shows the natural drop off in illumination upon entering the mine in terms of natural sunlight, which was roughly overhead and unable to illuminate far into the tunnel. By comparison, the 3D model produced a 3D rendering of the mouth of the tunnel without an overhead light source and uniform illumination of all the surfaces. The captured data provide a clear and accurate representation of the mine adit down to centimeter-scale features such as protrusions and fissures.

The inspection of the orthophoto mesh data further into the mine shows that the uranyl fluorescence of the minerals on the tunnel wall surfaces is clearly discernible. The green–yellow light that is emitted from the minerals is distinct in color and denotes the presence of the uranyl. The captured fluorescence is sufficiently bright that, in those parts of the mine entrance that still received some illumination by sunlight, the uranyl fluorescence was still successfully captured, as presented in Figure 8. This shows that fluorescence mapping of minerals can be achieved in low-light conditions as well as darkness.

The captured RGB color map recorded in the dark areas of the mine displayed a non-uniform patterning with respect to purple light. This purple light originates from the blacklight and highlights how the LEDs responsible for generating the UV light have an emission tail that falls into the visible spectrum. The variation and intensity regarding the captured purple light indicate how the manual scanning technique used to capture the data was not able to maintain a uniform stand-off distance from all the surfaces. In locations where the blacklight was held physically closer to the surface, the corresponding part of the orthophoto mesh that was captured exhibited a more intense purple coloration.

The ‘tramp’ purple light produced by the blacklight proved to be operationally useful,

enabling the operator to see the immediate area of the mine being scanned;and providing localized illumination of the rock surfaces, meaning that the recorded orthophoto captured textural detail, showing the physical structure of the exposed rock strata.

This extra structural information is potentially very useful for assisting in precisely locating uranyl mineral occurrences. This is clearly depicted in Figure 10, which shows a representation of the captured 3D fluorescence mesh, looking backwards towards the mouth of the mine tunnel and showing an area of the tunnel wall where discrete deposits of uranyl minerals have been captured with good spatial resolution and clarity, down to centimeter-scale lengths.

### 3.2. Spectral Analysis of UV-Induced Fluorescence from Uranyl-Bearing Mineral Samples

#### 3.2.1. Experimental Setup

Regarding the spectral analysis of Figure 6, the uranium mineral fluorescence was performed using a visible light spectrometer (SR-193i-A-SIL, Oxford Instruments Andor, Belfast, UK) coupled with a CCD (Charge-Coupled Device, DU420A-BEX2-DD, Oxford Instruments Andor, Belfast, UK). The UV torch (DARKBEAM B40 PRO MAX, Darkbeam, Shenzhen, China), with a central emission wavelength of 365 nm, was used to induce fluorescence. During the experiment, the UV torch was positioned 10 cm from the uranyl-bearing sample. The experimental setup is illustrated in Figure 12a.

To minimize background light interference, a filter system was developed, as shown in Figure 12b. Filter 1 is a Hoya U340 (UQG Ltd., Cambridge, UK), a UV-transmitting filter with a bandpass region of 275–375 nm [33]. It can block visible and infrared emissions from the UV torch. To enhance the blocking efficiency, three stacked filters were employed. The UV light from the torch passes through Filter 1 and induces fluorescence in the sample. Filter 2 is an FF01-550/200-25 visible-transmitting filter (Laser 2000 Photonics, Cambridge, UK) with a bandpass region of 450–650 nm. It can block the UV light from the torch and residual infrared light, allowing only the visible fluorescence signal to reach the spectrometer for further analysis [34].

#### 3.2.2. Data Analysis and Results

The recorded spectral data were converted to ASC output files using Andor SOLIS software (version 4.30.30024.0). The final fluorescence spectrum was obtained by subtracting the background data from the signal data using Python 3.11.9, with an example fluorescence spectrum shown in Figure 13. Five peaks were identified at 501.8 nm, 522.5 nm, 545.8 nm, 570.7 nm, and 598.5 nm, with corresponding signal intensity ratios of 9:24.3:13:4.7:1.

When compared with the data presented in Figure 1, the minerals collected in this study are characteristic of uranyl phosphates.

### 3.3. Raman Analysis

Representative Raman spectra collected from Figure 6 are shown in Figure 14. The raman spectra identified a strong uranyl symmetric stretch observed at 844 cm^−1^ and 1004 cm^−1^. Low-intensity bands situated at 188 and 617 cm^−1^ are also seen that are expected for the uranyl bending modes. Sulfate stretching modes are present in the spectra between 1000 and 1100 cm^−1^.

Raman bands were also observed at wave numbers of 312, 423, 457, 625, 830, 861, 978, and 995 cm^−1^. The 300–1000 cm^−1^ region in the spectrum emphasises the torbernite mineral phase [35,36]. Raman bands at 978 and 830 cm^−1^ are attributed to the $\nu_{3}$ antisymmetric stretching modes of ${\left({\mathrm{UO}}_{2}\right)}^{2+}$ units. The most intense bands in the Raman spectrum are observed at 861 ${\mathrm{cm}}^{-1}$ and 978 ${\mathrm{cm}}^{-1}$. These bands are assigned to the phosphate ${\mathrm{PO}}_{4}$$\nu_{3}$ antisymmetric stretching modes [36].

A band of low intensity located at 1050.8 ${\mathrm{cm}}^{-1}$ is also observed, which is assigned to a carbonate symmetric stretching mode ($\nu_{1}$) and is ascribed to carbonate minerals, possibly to andersonite (Na_2_Ca(UO_2_)(CO_3_)_3_ · 6 H_2_O) or schröckingerite (NaCa_3_(UO_2_)(CO_3_)_3_(SO_4_)F · 10 H_2_O) [35].

### 3.4. Elemental Composition Determined via Energy-Dispersive X-Ray Analysis

The elemental analysis of Figure 6 was conducted using electron microscopy. Uranyl minerals were initially identified by using backscattered electron imaging. The high-Z-number elemental content of the uranyl minerals made them easy to discern versus common silicate minerals. Both area and spot analyses of mineral occurrences were performed using energy-dispersive X-ray analysis. The recorded data indicated that the uranyl minerals were composed of significant quantities of sulfur and phosphorus, corresponding with the mineral attributions from the Raman and fluorescence analyses.

The surrounding mineral phases were typically bladed and elongate, with elemental compositions that are typical of magnesium and aluminium sulfates.

## 4. Ground Truth Mineral Sample Analysis

A materials analysis of the mineral samples collected from the mine, based on their exhibited fluorescence, provided confirmation that the samples were indeed uranium-containing. The elemental analysis indicated that the uranium-bearing mineral phases were primarily sulfate-related, which was not unexpected given the pyrite and chalcopyrite content of the surrounding rock formation. Lesser amounts of phosphorus were also detected, which equally has an association with ancient sulfide mineralization. A spot analysis on these mineral phases using Raman spectroscopy provided chemical confirmation that the mineral species contained uranyl. This was evidenced by the pair of uranyl-related vibrational peaks recorded at wave numbers of 978 and 830 ${\mathrm{cm}}^{-1}$, cross-correlating with library data for uranyl minerals of similar elemental composition. This chemical confirmation provides further strengthening of the assertion that the mineralization exhibiting the recorded green-light fluorescence was indeed uranyl-based. The combined electron microscopy, fluorescence, and Raman analysis indicates that the uranyl mineralization at this particular mine was predominantly johannite ($\mathrm{Cu}{\left({\mathrm{UO}}_{2}\right)}_{2}\left({\mathrm{SO}}_{4}\right){\left(\mathrm{OH}\right)}_{2}\;\cdot \;8\;H_{2}O$).

A spectral analysis of the of the observed uranium fluorescence corresponded closely to the previously documented fluorescence wavelengths of other uranyl mineral occurrences. The correlation regarding the emitted wavelength was not exact, but this was not unexpected because the exact color (wavelength) of the visible light emitted from a fluorescent mineral depends on many factors, notably the UV illumination wavelength, the crystal structure of the mineral, and the types of impurities present within the structure. From the 4200 mineral species documented, about 566 (7.4%) have been categorized as fluorescent [37,38], but, within a single fluorescent mineral type, there can be notable variations in emitted light between the end-member compositions.

It should be acknowledged that the exposed rock surfaces within the mine are constituted of many different mineral types and, as stated in the introduction, include other minerals previously reported for this mine, which naturally also exhibit UV-induced fluorescence. While all minerals can reflect light, fluorescent minerals absorb UV and convert it into longer wavelengths before emitting it back. Fortunately, these other fluorescent minerals do not emit at the same wavelength as observed directly for the uranyl minerals. For example, UV-induced fluorescence of calcite produces light that is wine-red in color, whilst gypsum emits broadly as white or white–blue light.

It should be noted that the uranyl minerals observed and mapped in this study do not represent the primary uranium mineralization found in the surrounding host rocks. The observed minerals may be considered as evaporation-related secondaries. The concept for the formation of these ‘secondary’ minerals is based on leaching of the surrounding sedimentary host rocks by meteoric waters. The surrounding host rocks contain horizons with higher organic content containing disseminated primary uranium minerals, assumed to be uraninite or pitchblende. Meteoric fluids that periodically fall on the mountain terrain, percolate through the sedimentary strata, and then, when reaching the organic and clay-rich horizons, flow laterally through the rock courses, due to the relative impermeability of these horizons and during their transit, act to partially dissolve the primary uranium minerals, as demonstrated in Figure 15. This leaching results in groundwater that is pregnant with uranyl and other associated metal salts from concurrent dissolution of other minerals. The electron microscopy data clearly indicate that sulfur and phosphorus are major components of the forming evaporites, indicating that they are also major elemental constituents of the surrounding country rock.

When the transiting ground waters meet the walls of the mine tunnel, the fluids are exposed to dry air and evaporation-caused precipitation of mineral salts containing uranium. The fact that the secondary uranyl minerals are observed in the mine indicates that the surrounding country rocks contain relatively substantial amounts of uranium. Therefore, the presence of these minerals, identified by their fluorescence, can be taken as an indicator that their parent minerals are in close physical proximity in the surrounding rocks, and that these parents occur in relatively high abundance—potentially at levels that are worthy of exploratory prospecting.

Based on this rationale, it is reasonable to consider that uranyl mineral fluorescence could be used as a means of rapidly screening legacy mines that were not previously exploited for uranium to determine if they would be worth revisiting as potential economic sources of uranium.

## 5. Discussion

Based on the data presented herein, the three-dimensional survey methodology is very promising but clearly not yet fully refined. In the future, it may be possible to further improve upon the quality of the data via several means.

Firstly, it would be appropriate to employ a UV illumination source that is brighter and/or has a larger field of illumination (wider solid-angle). In the present work, the UV light source produced small quantities of light in the visible spectrum as purple light, which proved to be very useful in terms of recording additional rock structure information via the RGB cameras, which, in postprocessing, made it possible to understand specifically where the fluorescing uranyl minerals were precisely located within the adit tunnel relative to specific rock structures and fancies. The tramp purple light was also useful in the acquisition of the data by providing sufficient visible illumination for the operative to physically see obstructions and features within the adit tunnel.

The adit surveyed in the current work has a linear geometry and provides a useful starting point in terms of developing and optimizing an effective data collection technique, which can be applied to more complex mine layouts.

During repeated experiments, it was determined that moving the camera too rapidly resulted in the LiDAR losing its fix, e.g., desynchronization, necessitating a full restart of the data collection. This led to the development of a methodical data collection strategy wherein the camera and coincident torch were swept slowly in arc-like movements to ensure full coverage of the exposed rock surfaces. It was also determined that, to avoid shadowing caused by rock protrusions, the survey path into the mine had to be retraced whilst facing the opposite direction.

It is expected that more complex mine layouts, whilst taking longer to survey, would have fewer incidences where localization is lost due to the presence of more distinctive texture and geometry features. Further testing is required to validate this working theory.

The established alternative approach to the scanning technique presented here would be the use of a LiDAR scanner mounted on a tripod, wherein a series of overlapping fixed-position scans starting from a known external datum point would occur throughout the mine. Such approaches may have higher specific resolutions but are comparably much slower and highly susceptible to shadowing issues due to the spacing of the data collection positions.

Further enhancements could possibly also be delivered through the use of robotic platforms to conduct the surveys instead of humans, e.g., quadruped robots or even UAVs. Removing the human operative from entering the mine workings and switching to a robotic platform would be intrinsically safer because human operators would not be subject to unknown amounts of gamma exposure, radon, as well as physical hazards, including mine collapse. However, utilizing robotic systems instead of humans is associated with notable deployment challenges. The example provided here in the tunnel was relatively straight, and the floor of the tunnel was relatively flat with relatively few obstructions. This would represent a very ‘easy’ deployment for robotic systems, whereas, in mines that are spread over multiple levels or with branching and bending tunnels, it would be a substantially greater challenge to achieve successful remote robotic deployments. However, as developments in the field of robotics and AI continue to advance, the ‘field’ capability of robotic platforms in unknown or unstructured environments might reasonably be expected to improve. Conceptually, such technological advancement would enable a human operator to oversee the survey but as a pilot or even remote supervisor who works from a safe location and not inside the mine.

This novel survey methodology clearly demonstrates and proves the concept for using combined solid-state scanning LiDAR with RGB photogrammetry to accurately map the 3D distribution of uranyl minerals on rock surfaces within legacy mines. With the duration of the full survey being only ≈20 min, this work shows how rapidly other mining sites could be comprehensively surveyed to determine the presence of uranyl minerals but also to capture the layout and 3D structure of the workings for the purposes of planning long-term management and decommissioning activities as well as modeling waterflow for pollution evolution assessments.

With the global renaissance in nuclear energy as a reliable energy alternative to fossil fuels, there is correspondingly an increase in interest for uranium production as a feedstock material for nuclear fuels. In recent months and years, the price of uranium has increased [39], and, accordingly, for legacy mines that were exploited over a half century ago (or longer), with poor residual documentation, there is an opportunity to revisit some of these defunct mine sites to appraise them for residual uranium resources or coincident valuable metals, such as W, Cu, Au, and REEs. For example, a tin mine that was operated in the early 1800s might well be revisited from a resource perspective by using rapid mapping techniques, as demonstrated here, to ascertain whether a more in-depth and costly resource assessment is worthwhile. It is notable that the tungsten-bearing mineral scheelite (CaWO_4_) exhibits UV-induced fluorescence, emitting in the blue, and could be surveyed in the same way as demonstrated for uranyl minerals here. Already, researchers have identified that drone-based surveying might be used to find scheelite deposits [40], and, in the past, fluorescence has been used to locate new mineral resources of tungsten [41].

## 6. Considerations for Future Work

Further work should test the accuracy of this data capture methodology, specifically the LiDAR scanning and capture capability. Test deployments in well-documented mines or tunnel-like facilities that constitute multiple levels or branching tunnels will ascertain whether the hardware and associated software are able to continuously maintain registry and correct orientation or whether significant drift will manifest in the data over long capture times.

Work should also examine the possible use of this fluorescence imaging technology in open-air applications for uranyl imaging. This is complementary to other authors’ prior work on surveying for scheelite [40,41] (CaWO_4_) with the light source and coincident camera/scanning system being deployed from the ground via booms or from the air via UAS (drone) platforms, with data capture occurring at night. In either case, the opportunity exists to conduct rapid assessments of legacy tailing piles for the presence of uranyl minerals.

## Figures and Tables

**Figure 1 sensors-25-02094-f001:**
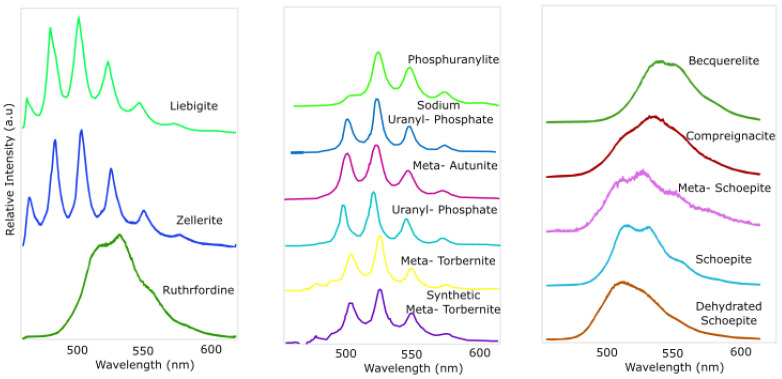
Fluorescence spectra adapted from [19] for uranium minerals at room temperature. All spectra were normalized to the same maximum intensity and offset along the Y-axis. The excitation wavelength, $\lambda_{\mathrm{ex}}$, used by [19] was 415 nm at the boundary of the violet spectral region.

**Figure 2 sensors-25-02094-f002:**
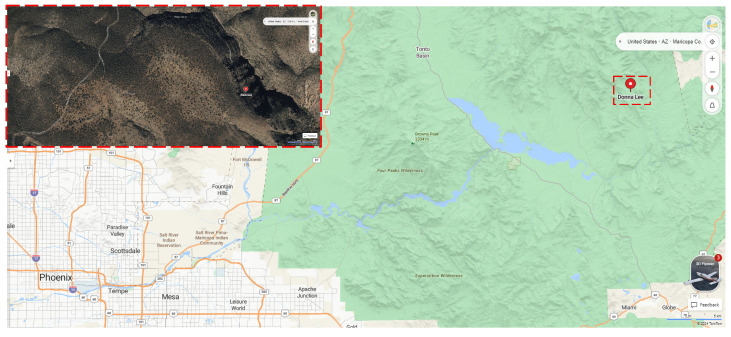
Location of the Donna Lee copper–uranium mine surveyed in this study.

**Figure 3 sensors-25-02094-f003:**
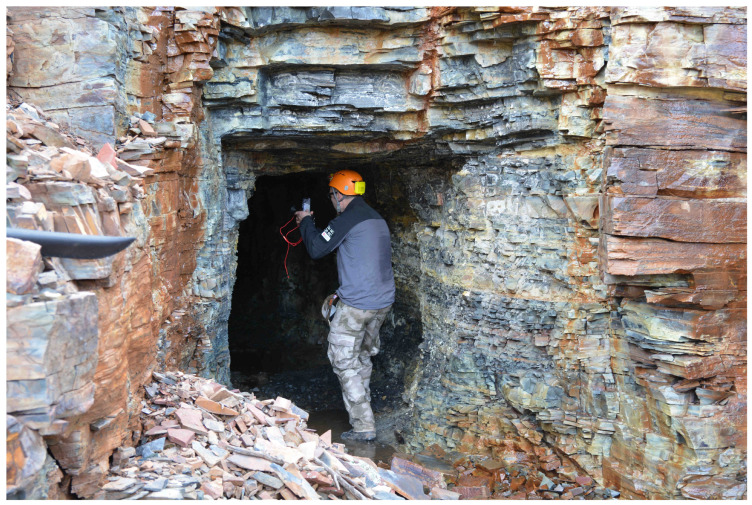
A surveyor entering the Donna Lee adit entrance, carrying the handheld survey apparatus.

**Figure 5 sensors-25-02094-f005:**
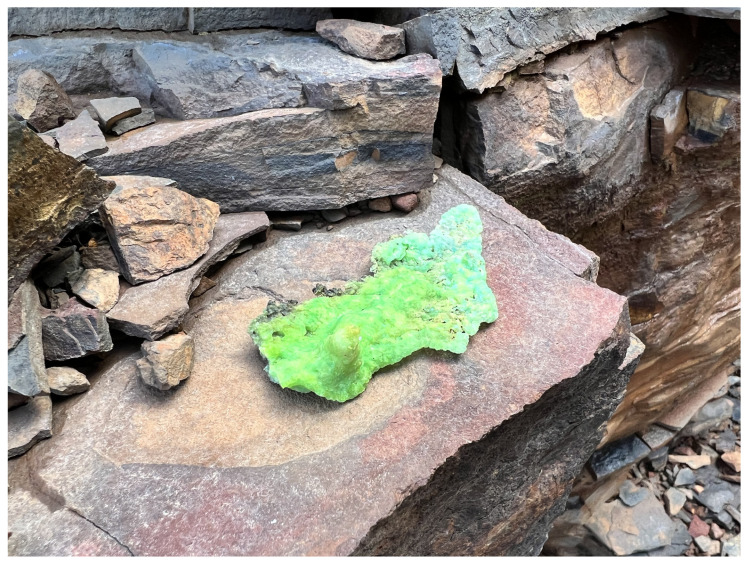
An example uranyl-bearing mineral sample that was collected from the mine.

**Figure 6 sensors-25-02094-f006:**
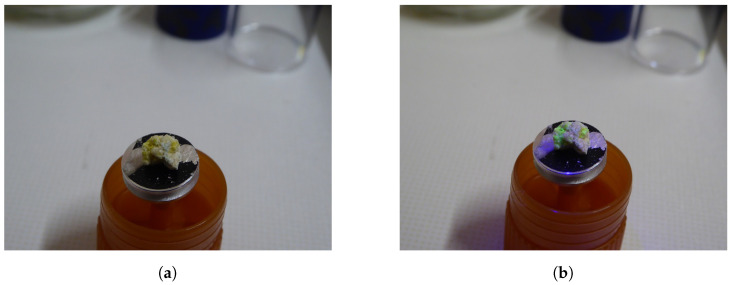
Laboratory sub-sample, mounted on an SEM-stub viewed under (**a**) natural light and (**b**) UV light.

**Figure 7 sensors-25-02094-f007:**
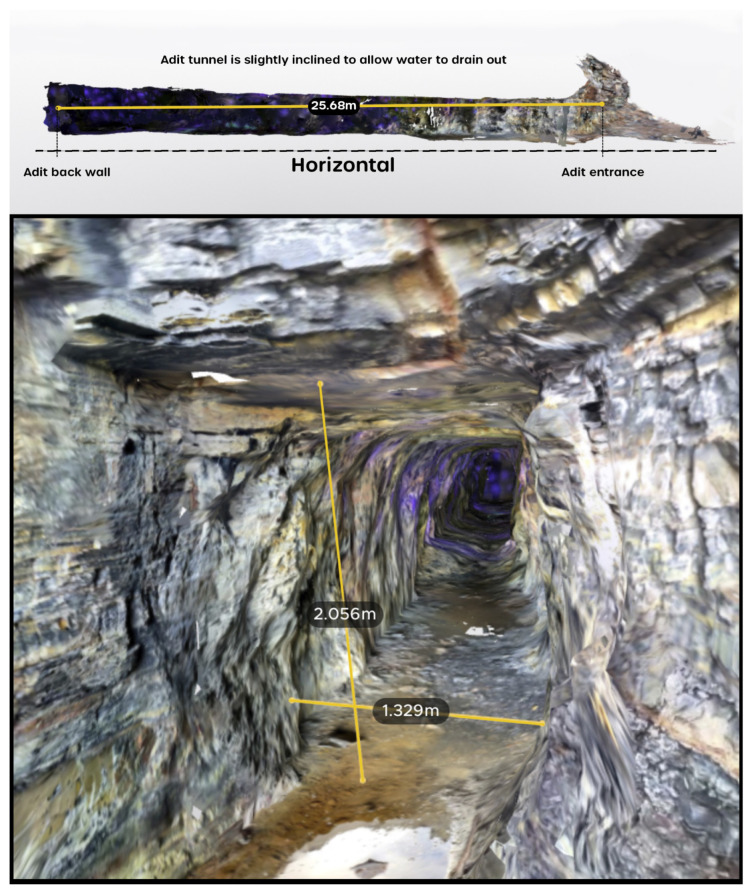
(**Top**) A cross-section of the mine adit showing its gentle upwards slope from the entrance, and (**bottom**) a render of the adit approximately 2m inside the entrance, showing the general width and height of the tunnel.

**Figure 8 sensors-25-02094-f008:**
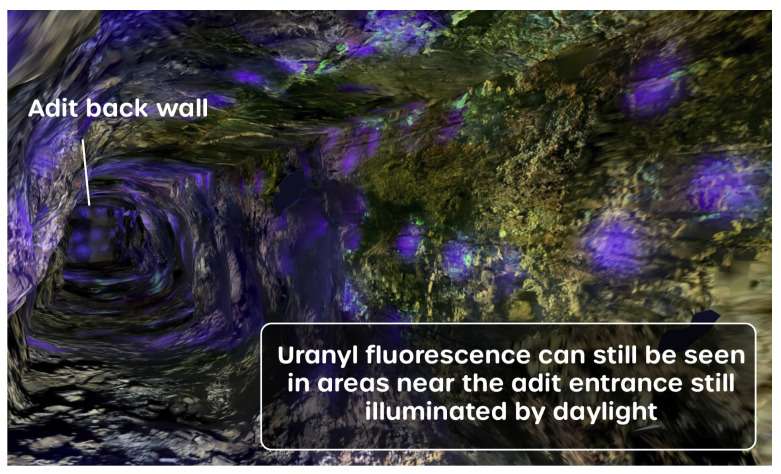
A render of the 3D model approximately 10m into the mine tunnel, showing that areas of green uranyl fluorescence could still be resolved in areas near the adit entrance still illuminated by daylight.

**Figure 9 sensors-25-02094-f009:**
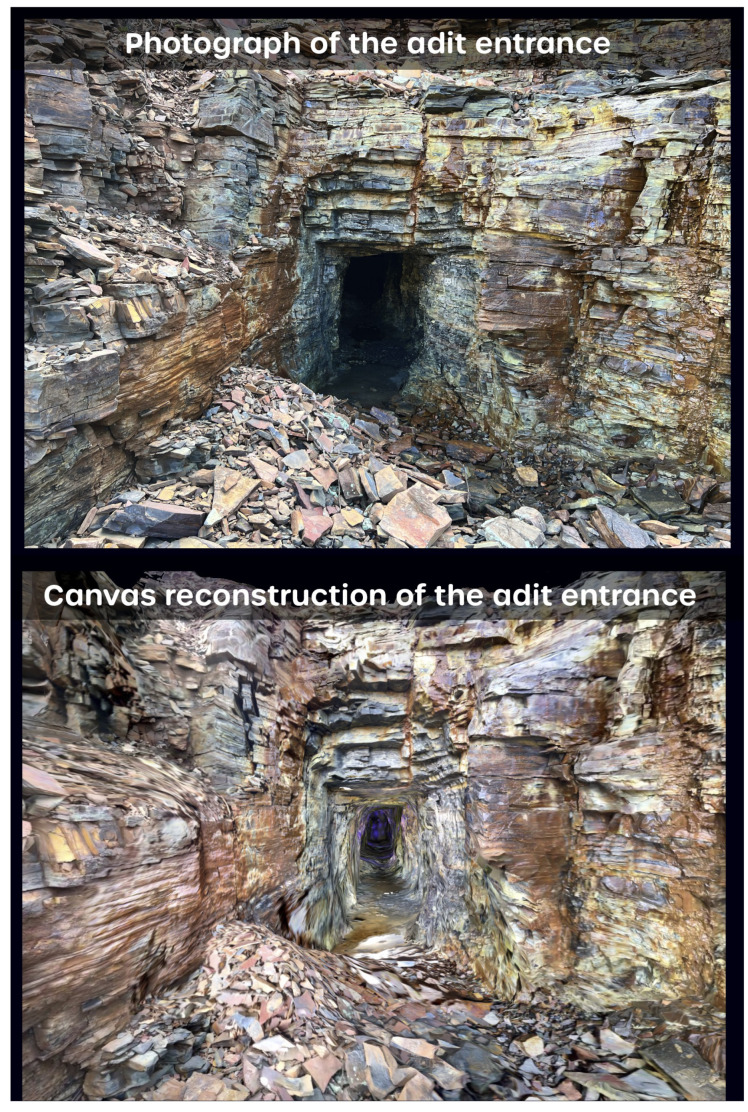
A side-by-side comparison of the real adit via photograph (**top**) versus the model reconstruction (**bottom**) produced by the Canvas software.

**Figure 10 sensors-25-02094-f010:**
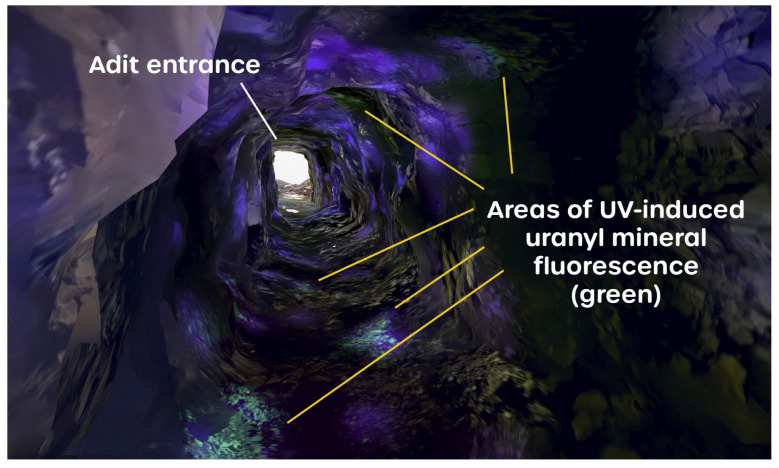
A render of the 3D model from inside the mine tunnel looking towards the mine entrance, showing areas of UV-induced uranyl mineral fluorescence in green.

**Figure 11 sensors-25-02094-f011:**
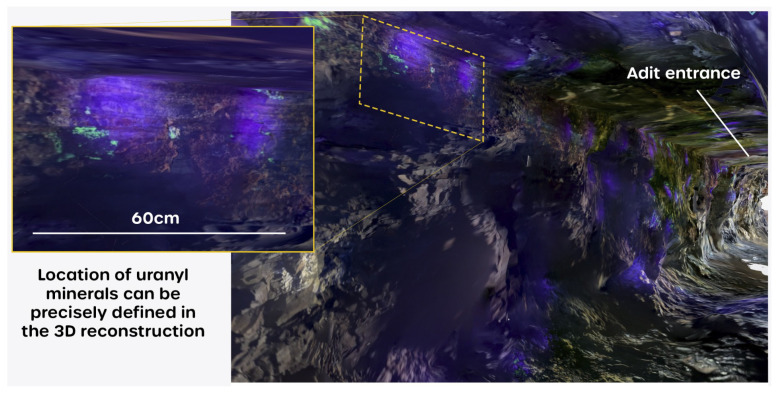
A section of the tunnel exhibiting uranyl mineral occurrence with (inset) a high-resolution image of the mineralization, showing how precisely the mineral outcrops can be defined in terms of shape and location.

**Figure 12 sensors-25-02094-f012:**
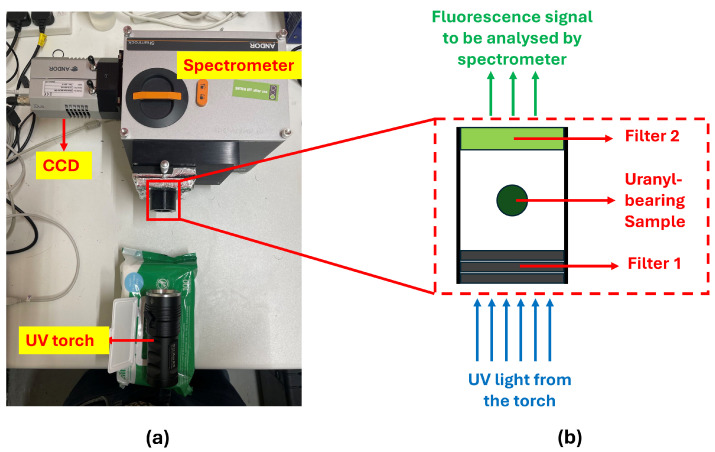
(**a**) Experimental setup for spectrum analysis of UV-induced fluorescence from a uranyl-bearing sample. (**b**) Filter system used for background light reduction.

**Figure 13 sensors-25-02094-f013:**
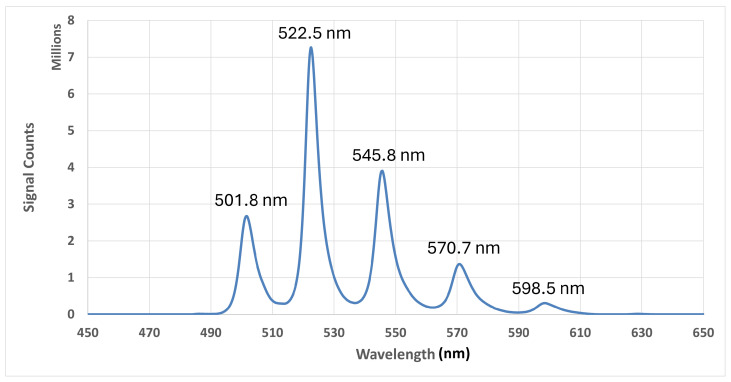
Fluorescence spectrum of the UV-induced uranyl-bearing sample.

**Figure 14 sensors-25-02094-f014:**
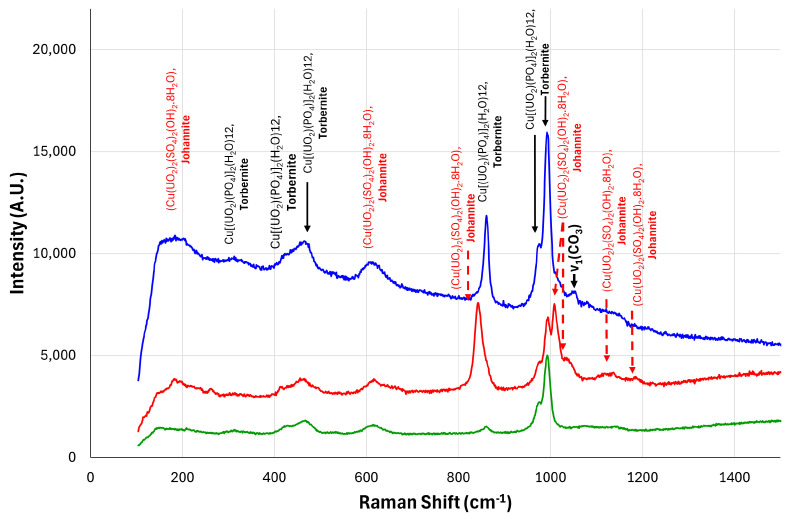
Annotated Raman spectra indicating the presence of two uranium-bearing minerals: torbernite and johannite. The three colors represent three different samples from three different locations in the mine.

**Figure 15 sensors-25-02094-f015:**
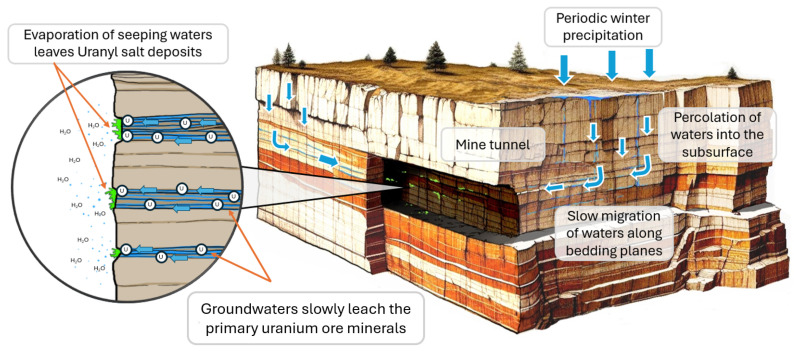
A graphical representation of the natural processes responsible for the formation of the uranyl minerals on exposed rock surfaces inside the mine.

**Table 1 sensors-25-02094-t001:** Properties and fluorescence peaks of selected uranyl minerals adapted from [17]. Fluorescence peak values from Fluomin.org (https://www.fluomin.org/fr/guidespectre.php (accessed on 17 January 2025)) [20,21,22].

Mineral	Composition	Crystal Structure	Fluorescence (Intensity/Color)	Fluorescence Peaks (nm)
Autunite	${\mathrm{Ca}(\mathrm{UO}}_{2}{)}_{2}{\left({\mathrm{PO}}_{4}\right)}_{2}\;\cdot \;10–12\;H_{2}O$	Tetragonal	Strong/Yellow–green	488.6, 504, 524.2, 548, 573.9, 602.4
Meta-autunite	${\mathrm{Ca}(\mathrm{UO}}_{2}{)}_{2}{\left({\mathrm{PO}}_{4}\right)}_{2}\;\cdot \;8\;H_{2}O$	Tetragonal	Strong/Yellow–green	488, 501.6, 523.7, 547.2, 573.3, 600.2
Metauranocircite	${\mathrm{Ba}(\mathrm{UO}}_{2}{)}_{2}{\left({\mathrm{PO}}_{4}\right)}_{2}\;\cdot \;8\;H_{2}O$	Tetragonal	Strong/Yellow–green	488.9, 502.5, 523.7, 547.4, 573.4, 602.6
Saléeite	${\mathrm{Mg}(\mathrm{UO}}_{2}{)}_{2}{\left({\mathrm{PO}}_{4}\right)}_{2}\;\cdot \;8–10\;H_{2}O$	Tetragonal	Strong/Yellow–green	535.2, 557.2, 579.3, 602.4, 625.6
Uranospinite	${\mathrm{Ca}(\mathrm{UO}}_{2}{)}_{2}{\left({\mathrm{AsO}}_{4}\right)}_{2}\;\cdot \;10\;H_{2}O$	Tetragonal	Strong/Yellow–green	501, 522, 546, 570, 597
Zippeite	${(\mathrm{UO}}_{2}{)}_{2}\left({\mathrm{SO}}_{4}\right){\left(\mathrm{OH}\right)}_{2}\;\cdot \;4\;H_{2}O$	Orthorhombic	Medium/Green	505, 527, 551, 577
Schroeckingerite	${\mathrm{NaCa}}_{3}{\left({\mathrm{UO}}_{2}\right)}_{3}{\left({\mathrm{CO}}_{3}\right)}_{3}$(SO_4_)F · 10 H2O	Orthorhombic	Strong/Blue–green	465, 483, 504, 526, 550, 577, 608, 641, 676, 713
Liebigite	${\mathrm{Ca}}_{2}\left({\mathrm{UO}}_{2}\right){\left({\mathrm{CO}}_{3}\right)}_{3}\;\cdot \;10\;H_{2}O$	Orthorhombic	Strong/Blue–green	469.8, 482.8, 503.3, 524.8, 548.5, 574.8, 604.6
Andersonite	${\mathrm{Na}}_{2}\mathrm{Ca}\left({\mathrm{UO}}_{2}\right){\left({\mathrm{CO}}_{3}\right)}_{3}\;\cdot \;6\;H_{2}O$	Rhombohedral	Strong/Blue–green	472, 483, 504, 526, 550, 577, 607, 640
Uranophane	${\mathrm{Ca}(\mathrm{UO}}_{2}{)}_{2}{\left({\mathrm{SiO}}_{3}\right)}_{2}{\left(\mathrm{OH}\right)}_{2}\;\cdot \;5\;H_{2}O$	Monoclinic	Medium/Green	499, 513, 535, 560, 586, 597, 609
β-Uranophane	${\mathrm{Ca}(\mathrm{UO}}_{2}{)}_{2}{\left({\mathrm{SiO}}_{3}\right)}_{2}{\left(\mathrm{OH}\right)}_{2}\;\cdot \;5\;H_{2}O$	Monoclinic	Weak/None/ (Yellow–green)	506, 527, 548, 565
Tyuyamunite	${\mathrm{Ca}(\mathrm{UO}}_{2}{)}_{2}{\left({\mathrm{VO}}_{4}\right)}_{2}\;\cdot \;5–8\;H_{2}O$	Orthorhombic	Weak/None/ (Yellow–green)	504, 526, 547, 573
Carnotite	$K_{2}{\left({\mathrm{UO}}_{2}\right)}_{2}{\left({\mathrm{VO}}_{4}\right)}_{2}\;\cdot \;1–3\;H_{2}O$	Monoclinic	Not fluorescent	Not fluorescent
Torbernite	${\mathrm{Cu}(\mathrm{UO}}_{2}{)}_{2}{\left({\mathrm{PO}}_{4}\right)}_{2}\;\cdot \;10\;H_{2}O$	Tetragonal	Not fluorescent	Not fluorescent
Meta-torbernite	${\mathrm{Cu}(\mathrm{UO}}_{2}{)}_{2}{\left({\mathrm{PO}}_{4}\right)}_{2}\;\cdot \;8\;H_{2}O$	Tetragonal	Not fluorescent	Not fluorescent

## Data Availability

The data presented in this study are available on request from the corresponding author due to the sensitive nature of the associated uranium-containing materials.

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
