# Peer review of "Combined Solid-State LiDAR and Fluorescence Photogrammetry Imaging to Determine Uranyl Mineral Distribution in a Legacy Uranium Mine"

_sensors, 2025, doi:10.3390/s25072094_

Round 1

Reviewer 1 Report

Comments and Suggestions for Authors

To accurately determine the chemical composition of rocks, serious laboratory research is needed in the field or laboratory conditions, which is associated with significant costs of human, instrumental and financial resources. Therefore, preliminary exploration of the chemical and mineral composition of mine ores on site is important in order to know exactly where to throw the main forces. Determination of uranium-containing ores in old abandoned mines became especially important, when no one had the slightest idea about uranium. Rainwater can carry uranium-containing compounds into groundwater, which is dangerous from the point of view of ecology and human health. In this paper, the authors propose a simple and inexpensive way to survey abandoned mines for the content of the mineral uranyl in them using “improvised” devices in the form of an ultraviolet flashlight and an iPhone 12 Pro smartphone with its standard equipment (lidar and three RGB cameras). As the authors have shown, the exact elemental composition is determined in laboratory conditions. However, the presence and spatial distribution of uranium-containing minerals can be determined with high accuracy using the approach they propose and with little effort. The authors have completed the entire cycle of work from the preliminary detection of uranium-containing minerals in the mine to their spectroscopic analysis.

The article undoubtedly deserves attention and falls within the scope of “Sensors”. I believe that the results of their work will be of interest to both specialists and the general public. The manuscript can be published, but some revisions are needed. Some of my comments are included below, while all of the comments can be found in the highlighted version of the manuscript attached to this review.

Some specific comments

Abstract

Line 11. I have some doubts about the correctness of using the term “three-dimensional distribution”. Such a distribution, in my opinion, implies the distribution of uranyl minerals inside the rock, and this cannot be done using the methods proposed by the authors. Perhaps it would be more correct to use the term “surface distribution” (inside the mine). However, I do not insist on this.

1. Introduction

1.1. Techniques for rapid surveying of radiological material

Line 97. It is a bit illogical to introduce Subsection 1.1 here (it would make sense if there were at least 2 subsections). As far as I know, the Introduction should end with an introduction to the problem and the objectives of this study, but not with descriptive information. I would recommend placing the entire Subsection 1.1 (without the number 1.1 and title) before the paragraph “Current work focuses on demonstrating the use of...” (see lines 86-96).

Lines 118–120. Quotation: “Fortunately the rapid identification and visualization of uranyl minerals can be achieved by using ultraviolet (UV) light illumination to stimulate a strong characteristic fluorescence.”

Comment: Well, ultraviolet is a loose concept. I would like to know what range of ultraviolet light used to illuminate uranyl minerals we are talking about. It is clear that this range should be between 200 and 400 nm, but what wavelengths are usually used in portable devices (not just in your case)?

Lines 128–129. Quotation: “Figure 1 and Table 1 show that the majority of common uranyl minerals are observed to exhibit characteristic yellow-green fluorescence when illuminated by a UV source.”

Comment: First, it should be clarified here that the fluorescence spectra were obtained at an excitation wavelength of 415 nm. There is no explanation for the lambda designation in the caption to Figure 1. Second, I am afraid that 415nm wavelength is not ultraviolet. According to https://en.wikipedia.org/wiki/Violet_(color), it should be violet light (380–435 nm).

2. Materials and Methods

2.1. Location

Line 134. The title of Subsection 2.1 (Location) does not quite accurately reflect its content. In addition to the location, the chemical composition of the ores present is also presented here.

Page 4 (Figure 1). Figure 1 taken from work [19] contains parts (a), (b), and (c). Therefore, either the designations of the parts of the figure must be removed or they must be described in the caption to the figure.

Page 4 (Table 1). There is already a title at the top of Table 1. If the authors want to use their own title, then the title from source [17] should be removed, or Table 1 should be classified as a figure taken entirely from [17]. The font size of the contents of Table 1 could be better increased by increasing the scale of Table 1 as a figure.

2.2.2. Fluorescence survey

Lines 173–174. Quotation:The precise specifications of the LiDAR unit have not been released by Apple, but the performance of the sensor has been extensively evaluated.”

Comment: Is the wavelength of the emitted radiation source of this lidar known, for example, from literary sources? If not, did the authors try to determine this wavelength, for example, using a monochromator?

Lines 181–182. Quotation: “Evaluations for indoor mapping have found an improved spatial accuracy to 3 cm when modeling a room 22 × 6m in size [26].”

Comment: The paper [26] presents Apple iPhone 13 Pro LiDAR accuracy assessment, while the authors use Apple iPhone 12 Pro. Do these two models use the same lidar?

2.2.4. Collection of ground truthing samples

Page 8 (Figure 6). Figure 6 is not mentioned anywhere in the text of the article.

3. Results

Lines 306–308. Quotation: “... discrete deposits of uranyl minerals have been captured with good spatial resolution and clarity, down to millimetre scale lengths.”

Comment: This is unlikely. With a measurement error of 3–4 cm, there can be no talk of millimeter-scale accuracy – only centimeter-scale accuracy.

3.1. Canvas Renders

Line 309. Subsection 3.1 is clearly out of place here. After it there are only Figures 7-11.

Page 11 (Figures 8 and 9). The sequence of Figures 8 and 9 is incorrect: Figure 9 comes before Figure 8.

3.2.1. Experimental Setup

Are the bandwidths of Filters 1 and 2 known? If so, please provide them here (after mentioning the relevant filter). It would also be nice to provide direct links to web pages describing the equipment mentioned, if possible.

3.2.2. Data Analysis and Results

Lines 331–333. Quotation:Four peaks were identified at 501.8nm, 522.5nm, 545.8nm, 570.7nm, and 598.5nm, with corresponding signal intensity ratios of 2.7 : 7.3 : 3.9 : 1.4 : 0.3.”

Comment: For some reason I counted 5 peaks. In my opinion, these ratios (2.7 : 7.3 : 3.9 : 1.4 : 0.3) are not exactly signal intensity ratios. They are simply specified data in millions of signal counts. To obtain signal intensity ratios, one of the values should be taken as a unit (for example, the smallest value), and the rest should be recalculated accordingly, i.e. 9 : 24.3 : 13 : 4.7 : 1.

3.3. Raman Analysis

Page 14 (Figure 14). First, please check the correctness of the chemical formula of torbernite (usually the number of physically attached water molecules comes before, not after, the chemical formula of water). Compare with the formula in lines 51–52 or in Table 1 of this manuscript. Second, all three colored curves (red, blue, and green) in Figure 14 should be explained in the caption to Figure 14 or the text of the manuscript.

4. Discussion

Line 362. In my opinion, Subsection “4.1. Ground truth mineral sample analysis” (together with Figure 15, lines 413–471) can hardly serve as part of the Discussion, but can be considered as a separate Section (with the same title) before the Discussion (lines 362–412).

4.1. Ground truth mineral sample analysis

Line 427. There should be two sulphate ions (SO4)2 in johannite. Please correct this and compare with Figure 14.

Page 17 (Figure 15). Figure 15 is not mentioned anywhere in the text of the article.

5. Considerations for future work

Lines 480–481. Quotation: “This is complementary to prior work on surveying for scheelite (CaWO4)...”

Comment: What is this work? Is there a reference to it?

References

Please check all references. For example, references [2, 27, and 28] are not formatted correctly.

Author Response

We have addressed all the reviewer comments (please see attached file which contains our responses to the comments of ALL the reviewers). We would like to thank the reviewer for their valuable observations and suggestions.

Text

Comment

Reply

Reviewer 1

Abstract

/

/

All grammatical corrections suggested by reviewer 1 have been implemented.

Line 11. I have some doubts about the correctness of using the term “three-dimensional distribution”. Such a distribution, in my opinion, implies the distribution of uranyl minerals inside the rock, and this cannot be done using the methods proposed by the authors. Perhaps it would be more correct to use the term “surface distribution” (inside the mine). However, I do not insist on this.

Now reads;

Such a simple methodology, presented

herein, can be used to quickly determine the surface distribution of uranyl minerals within the

underground workings and provide an indication of the presence of primary uranium ore minerals

buried within the surrounding rock.

1. Introduction

/

/

Line 97. It is a bit illogical to introduce Subsection 1.1 here (it would make sense if there were at least 2 subsections). As far as I know, the Introduction should end with an introduction to the problem and the objectives of this study, but not with descriptive information. I would recommend placing the entire Subsection 1.1 (without the number 1.1 and title) before the paragraph “Current work focuses on demonstrating the use of...” (see lines 86-96).

The authors agree. The paragraphs have been moved to the suggested location above.

Lines 118–120. Quotation: “Fortunately the rapid identification and visualization of uranyl minerals can be achieved by using ultraviolet (UV) light illumination to stimulate a strong characteristic fluorescence.”

Comment: Well, ultraviolet is a loose concept. I would like to know what range of ultraviolet light used to illuminate uranyl minerals we are talking about. It is clear that this range should be between 200 and 400 nm, but what wavelengths are usually used in portable devices (not just in your case)?

The lines have been amended to make the point clearer. We have also added the UV wavelength 365nm.

Lines 128–129. Quotation: “Figure 1 and Table 1 show that the majority of common uranyl minerals are observed to exhibit characteristic yellow-green fluorescence when illuminated by a UV source.”

Comment: First, it should be clarified here that the fluorescence spectra were obtained at an excitation wavelength of 415 nm. There is no explanation for the lambda designation in the caption to Figure 1. Second, I am afraid that 415nm wavelength is not ultraviolet. According to https://en.wikipedia.org/wiki/Violet_(color), it should be violet light (380–435 nm).

The authors agree with this comment.  It has been corrected to address this.

Table 1 shows the majority of common uranyl minerals are observed to exhibit characteristic yellow-green fluorescence when illuminated by a UV source. Figure 1 adapted from [ref] shows that this property can potentially be exploited, especially in dark underground workings, to quickly show if such minerals are present on exposed rock surfaces and if so, to map their exact location and distribution.

Lamba ex is the excitation wavelength that the study [ref] used. This 415nm is just into the violet spectrum.

2. Materials and Methods

/

Line 134. The title of Subsection 2.1 (Location) does not quite accurately reflect its content. In addition to the location, the chemical composition of the ores present is also presented here.

The authors agree with this comment. Section has been renamed to:

Lithological Location

Which better covers both the geographical and geological extent of the area.

Page 4 (Figure 1). Figure 1 taken from work [19] contains parts (a), (b), and (c). Therefore, either the designations of the parts of the figure must be removed or they must be described in the caption to the figure.

These figures were recreated and adapted from the source literature, rather than utilising them direct. This was sent to the editor during submission.   We believe that the author made table has already actioned the comment.

Page 4 (Table 1). There is already a title at the top of Table 1. If the authors want to use their own title, then the title from source [17] should be removed, or Table 1 should be classified as a figure taken entirely from [17]. The font size of the contents of Table 1 could be better increased by increasing the scale of Table 1 as a figure.

Similar response to above, where we remade and adapted the table from the source paper. For some reason this again was not distributed to the reviewer during submission. We believe that the author-made table has already addressed the comment.

Lines 173–174. Quotation: “The precise specifications of the LiDAR unit have not been released by Apple, but the performance of the sensor has been extensively evaluated.”

Comment: Is the wavelength of the emitted radiation source of this lidar known, for example, from literary sources? If not, did the authors try to determine this wavelength, for example, using a monochromator?

The wavelength of the LIDAR built-into the iPhone is ~940nm. Accordingly, we have modified the section to state this.

Lines 181–182. Quotation: “Evaluations for indoor mapping have found an improved spatial accuracy to 3 cm when modeling a room 22 × 6m in size [26].”

Comment: The paper [26] presents Apple iPhone 13 Pro LiDAR accuracy assessment, while the authors use Apple iPhone 12 Pro. Do these two models use the same lidar?

This was a text error in the manuscript. We used an iPhone 13 Pro for our survey, so are directly comparable to the other studies.

Page 8 (Figure 6). Figure 6 is not mentioned anywhere in the text of the article.

Now referenced under 2.2.5 as well as 3.2.1 , 3.2.2 and 3.4

3. Results

/

Lines 306–308. Quotation: “... discrete deposits of uranyl minerals have been captured with good spatial resolution and clarity, down to millimetre scale lengths.”

Comment: This is unlikely. With a measurement error of 3–4 cm, there can be no talk of millimeter-scale accuracy – only centimeter-scale accuracy.

We have amended the manuscript to state ‘centimetre accuracy’. Many thanks for catching this.

Line 309. Subsection 3.1 is clearly out of place here. After it there are only Figures 7-11.

The authors agree with this.

Move subsection 3.1 to the top under \results

Page 11 (Figures 8 and 9). The sequence of Figures 8 and 9 is incorrect: Figure 9 comes before Figure 8.

The Figures are now in Chronological order

Are the bandwidths of Filters 1 and 2 known? If so, please provide them here (after mentioning the relevant filter). It would also be nice to provide direct links to web pages describing the equipment mentioned, if possible.

Filter1 with bandpass region of 275 - 375 nm. Filter2 with bandpass region of 450 – 650 nm. The links are provided and cited.

Lines 331–333. Quotation: “Four peaks were identified at 501.8nm, 522.5nm, 545.8nm, 570.7nm, and 598.5nm, with corresponding signal intensity ratios of 2.7 : 7.3 : 3.9 : 1.4 : 0.3.”

Comment: For some reason I counted 5 peaks. In my opinion, these ratios (2.7 : 7.3 : 3.9 : 1.4 : 0.3) are not exactly signal intensity ratios. They are simply specified data in millions of signal counts. To obtain signal intensity ratios, one of the values should be taken as a unit (for example, the smallest value), and the rest should be recalculated accordingly, i.e. 9 : 24.3 : 13 : 4.7 : 1.

Modified to 5 peaks with ratio 9 : 24.3 : 13 : 4.7 : 1.

Page 14 (Figure 14). First, please check the correctness of the chemical formula of torbernite (usually the number of physically attached water molecules comes before, not after, the chemical formula of water). Compare with the formula in lines 51–52 or in Table 1 of this manuscript. Second, all three colored curves (red, blue, and green) in Figure 14 should be explained in the caption to Figure 14 or the text of the manuscript.

We have checked the document and we can confirm that we have stated the formula for meta-torbernite correctly in the main text. Meta-torbernite contains less water than torbernite, and therefore in this very arid environment meta-torbernite is considered the predominant mineral form.

4. Discussion

/

Line 362. In my opinion, Subsection “4.1. Ground truth mineral sample analysis” (together with Figure 15, lines 413–471) can hardly serve as part of the Discussion, but can be considered as a separate Section (with the same title) before the Discussion (lines 362–412).

The author agrees and it has been moved out of discussion

Line 427. There should be two sulphate ions (SO4)2 in johannite. Please correct this and compare with Figure 14.

This has been corrected.

Page 17 (Figure 15). Figure 15 is not mentioned anywhere in the text of the article.

Now Refd at end of sentence,

5. Considerations for future work

Lines 480–481. Quotation: “This is complementary to prior work on surveying for scheelite (CaWO4)...”

Comment: What is this work? Is there a reference to it?

References have been added [38,39] and the text amended.

References

Please check all references. For example, references [2, 27, and 28] are not formatted correctly.

The references have been checked as correct

Reviewer 2 Report

Comments and Suggestions for Authors

Dear authors,

thank you for the interesting research and application and combination of sensors to quickly and at low-cost determination of potential resources.

The aim is clearly defined and supported by the introduction and relevant references.

The methodology is well prepared, reference samples were analyzed by advanced technology down to the molecular level for mineral identification, as well as by spectral analysis.

I can support your findings regarding the accuracy of the mapping technology of 3-4 cm by measuring solid state lines at the time of the survey and in the 3D model as well as in the generated orthophoto of the entrance of the mine.

I think you provided a satisfactory explanation in the discussion regarding the formation of uranyl minerals on the rock surfaces.

I don't have any recommendations and I am looking forward for your further research in this field.

Reviewer 3 Report

Comments and Suggestions for Authors

  1. When the LIDAR sensor based on iPhone12 Pro collects data, need to provide more details about the parameter Settings? The scope, format, etc. of the acquired data need to be further provided.
  2. It is necessary to clearly state whether the LiDAR and camera of the iPhone 12 Pro were calibrated, as well as the calibration methods and frequencies, to ensure the accuracy of the data.
  3. Does the LiDAR sensor on the iPhone12 Pro need to be calibrated? What method of calibration is used? The effect of calibration on the collected data needs to be clarified.
  4. For the 3D model generated in the manuscript, there should be a detailed evaluation and verification process for its geometric positioning accuracy.
  5. The impact of environmental factors (such as temperature, humidity, ventilation, etc.) on the detection results was not fully discussed in the research. It is recommended to supplement relevant content.
  6. There are inconsistent problems in the citation formats of some references. It is necessary to unify and adjust them in accordance with the citation norms of the journal.
  7. The annotations of some charts are not detailed enough. More information, such as axis units and chart explanations, should be added to enable readers to better understand the chart content. As Figure 1 and Table 1 are not clear enough to read.

Reviewer 4 Report

Comments and Suggestions for Authors

Pleases see attachment

Round 2

Reviewer 4 Report

Comments and Suggestions for Authors

accept

Comments on the Quality of English Language

OK